# Four-year nationwide incidence of retinitis pigmentosa in South Korea: a population-based retrospective study from 2011 to 2014

Tyler Hyungtaek Rim,[1,2] Hye Won Park,[1,2] Dong Wook Kim,[3] Eun Jee Chung[1]

► Prepublication history and additional material are available. To view please visit the journal (http://dx.doi.org/10.1136/bmjopen-2016-015531).

THR and HWP contributed equally.

## ABSTRACT

**Objective** To determine the incidence of retinitis pigmentosa (RP) in South Korea.

**Design** Nationwide, population-based retrospective study.

**Setting** Census population of South Korea

**Participants** This study involved the entire population of South Korea (n=47 990 761). Patients confirmed as having RP by an ophthalmologist from 1 January 2011 to 31 December 2014 were included.

**Primary outcome measure** The average incidence of RP during the 4-year study period was estimated using population data from the 2010 Korean census.

**Results** A total of 3144 (1567 men and 1577 women) patients confirmed as having RP were identified. The average incidence of RP was 1.64 cases/100 000 person-years (95% CI 1.58 to 1.70). The incidence of RP distribution skewed to the left across age groups, with one smaller peak observed in the 20–24-year-old age group (1.24 cases/100 000 person-years) and a larger peak observed in the 65–69-year-old age group (3.26 cases/100 000 person-years). The overall incidence was similar in men and women (1.64 cases/100 000 person-years (95% CI 1.56 to 1.73) for men; 1.63 cases/100 000 person-years (95% CI 1.55 to 1.72) for women).

**Conclusions** Our study's estimates of the nationwide population-based incidence of RP in an Asian population will help advance the understanding of the disease onset and allow healthcare systems to plan accordingly.

### Strengths and limitations of this study

► In this retrospective cohort study of registry data on retinitis pigmentosa (RP) for the entire South Korean population, the incidence of RP was found to be 1.64 cases per 1 00 000 person-years.

► This study is unique because it included the largest population ever studied in terms of this rare disease, and it represents the first investigation of the nationwide incidence of RP in an Asian country.

► Only a few recent studies have collected data for a general population and calculated the incidence of RP. There are no nationwide RP-related epidemiological studies.

► A lack of clinical information, including family pedigree, visual acuity, visual field and genetic analysis, is an inherent limitation of the present claims database study.

► Although not ideal, our approach was cost-effective for identifying patients with RP and calculating the incidence of RP in the entire South Korean population.

[1]Department of Ophthalmology, National Health Insurance Service Ilsan Hospital, Goyang, Korea
[2]Department of Ophthalmology, Severance Hospital, Institute of Vision Research, Yonsei University College of Medicine, Seoul, Korea
[3]Department of Policy Research Affairs, National Health Insurance Service Ilsan Hospital, Goyang, Korea

**Correspondence to**
Eun Jee Chung;
eunjee95@nhimc.or.kr

## INTRODUCTION

Retinitis pigmentosa (RP) includes a group of heterogeneous hereditary retinal degeneration disorders caused by the progressive loss of photoreceptor cells.[1 2] It is characterised by widespread retinal degeneration and is an important and leading cause of blindness.[3] However, there is a paucity of data based on large, well-defined populations regarding the epidemiological characteristics of RP. Ideally, a study evaluating the epidemiology of RP should be based on population data rather than on data obtained from a tertiary referral centre to avoid the effects of a referral bias. To the best of our knowledge, only a few recent studies have collected data from a general population and calculated the incidence of RP. The incidence of RP was reported in only two previous studies conducted in Denmark (0.79 cases/100 000 person-years)[4] and in the state of Maine in the USA (6 cases/100 000 person-years).[5] Despite recent large-scale health-related studies,[6–8] there are no nationwide RP-related epidemiological studies. In general, incidence estimates provide more information than prevalence estimates with regard to a disease's characteristics and to help healthcare systems plan accordingly.

South Korea has a mandatory universal health insurance system covering the entire population of 48 million people; therefore, the medical claims database includes all healthcare use in South Korea. In addition, the Korean government initiated a registration programme for rare and intractable disorders, such as RP, in 2007. Patients with RP registered in this programme are eligible for

up to a 90% co-payment reduction after the diagnosis has been confirmed by an ophthalmologist using the Korean National Health Insurance Service (KNHIS) diagnostic criteria based on an electrophysiological examination. Our aim was to use this database to conduct a nationwide, population-based study to determine the incidence of RP in South Korea.

## SUBJECTS AND METHODS

### Statement of ethics

This retrospective nationwide cohort study was reviewed and approved by the Institutional Review Board of the National Health Insurance Service Ilsan Hospital, Gyeonggi-do, Korea. This study adhered to the tenets of the Declaration of Helsinki, and written informed consent was waived.

### RP registration in South Korea and its definition

All Korean residents are obligated to enrol in the KNHIS. Claims are accompanied by data regarding diagnostic codes, procedures, prescription drugs, personal information, and information about the hospital. No patient healthcare records are duplicated because all Korean residents receive a unique identification number at birth. Furthermore, the KNHIS uses the Korean Classification of Diseases, which is a system similar to the International Classification of Diseases. In 2007, the National Health Insurance Service initiated a co-payment reduction of up to 90% for patients suffering from 138 rare and intractable disorders, including RP. Patients with RP who registered in the programme were eligible for co-payment reduction after receiving a confirmed diagnosis by an ophthalmologist based on the National Health Insurance Service diagnostic criteria. The National Health Insurance Service diagnostic criteria for RP require: (1) ophthalmoscopic abnormalities of the retina on a dilated fundus examination; and (2) electroretinographic changes confirming the presence of RP-related photoreceptor damage. All submissions for co-payment reduction registration were reviewed and confirmed by the Health Insurance Review Agency. Patients must reapply for co-payment reduction registration every 5 years after the initial registration to maintain the co-payment reduction. After registration, all RP-related claims contain the RP registration code (V209) in addition to the diagnostic code for RP (Korean Classification of Diseases H3551). Patients who filed claims for RP (V209, H3551) from January 2011 to December 2014 with the KNHIS were included in this study. We excluded patients with chronic RP registered between 1 January 2007 and 31 December 2010, because RP (H3551) and Stargardt disease (H3558) shared the same registration code (V209) and did not have a Korean Classification of Diseases diagnostic code in the KNHIS database during this time. Finally, we obtained RP registration data from the national health claim database between 2007 and 2014, and calculated the incidence of RP between 2011 and 2014.

### Statistical analysis

The incident time was defined as the registration date of a rare and intractable disorder diagnosed as RP. Annual population data were obtained from the Population and Housing Census conducted in 2010 and available from the Korean Statistical Information Service (http://kosis.kr). Detailed demographic characteristics of the South Korean population are listed in table 1. The person-time incidence rates for 2011–2014 were calculated as the number of people who developed RP divided by the total person-time at risk during the study period. Therefore, in this analysis, person-years were counted after the incident time. The incidence per 100 000 person-years, based on the 2010 census, was estimated using the Poisson distribution. In our explorative analysis (see online Supplementary appendix) the total number of RP cases from 2007 to 2014 was estimated at 7424. Prevalent cases of RP were not excluded from the population at risk because the denominator (total Korean person-time at risk) was large enough to not be affected in terms of incidence. The total person-time at risk was assumed as the total population (census in 2010, n=47 990 761, table 1) over 4 years from 2011 to 2014 (population multiplied by four). The annual incidence from 2011 to 2014 and overall incidence were estimated with 95% confidence intervals; additionally, the age-specific and sex-specific incidence rates were estimated, and the age-specific population for each year was approximated based on the 2010 census population. The male-to-female ratio for the RP incidence rate was also estimated. A significance level of 0.05 was selected. All analyses were conducted using Stata/MP, version 14.0 (StataCorp, College Station, TX, USA).

## RESULTS

The total number of incident cases of RP from 2011 to 2014 in South Korea was 3144, including 1567 men (49.9%) and 1577 women (50.1%). The incidence of RP during the study period was 1.64 cases/100 000 person-years. In men the incidence was 1.64 cases/100 000 person-years, and in women it was 1.63 cases/100 000 person-years. The peak incidence in the total population was observed in the 65–69-year-old age group (3.26 cases/100 000 person-years; men 3.00 cases/100 000 person-years; women 3.86 cases/100 000 person-years) (table 1 and figure 1). Although there was some variation between the different periods and age groups, the overall incidences in men and women were similar (male-to-female ratio=1.01, p=0.80). The incidence of RP distribution was negatively skewed (skewed to the left) across age groups, with one smaller peak observed in 20–24-year-olds (1.24 cases/100 000 person-years). The incidence was slightly decreased in the 25–29-year-old age group (1.07 cases/100 000 person-years), and the maximum incidence was seen in the 65–69-year-old age group (3.26 cases/100 000 person-years) (figure 1). The incidence for women steadily increased with age, reaching a maximum level in the 65–69-year-old age group, and then it gradually decreased

**Table 1** Demographic characteristics of South Korea and the incidence of retinitis pigmentosa per 100 000 person-years in the South Korean population (2011–2014)

| Age (years) | Number of Koreans | | | Total | | Men | | Women | | Male-to-female Ratio | p Value ($\chi^2$ test) |
|---|---|---|---|---|---|---|---|---|---|---|---|
| | Total | Men | Women | No. | Incidence (95% CI) | No. | Incidence (95% CI) | No. | Incidence (95% CI) | | |
| 0–4 | 2 219 084 | 1 142 220 | 1 076 864 | 4 | 0.05 0.01 to 0.12 | 4 | 0.09 0.02 to 0.22 | 0 | No observation | NA | 0.05 |
| 5–9 | 2 394 663 | 1 243 294 | 1 151 369 | 21 | 0.22 0.14 to 0.34 | 14 | 0.28 0.15 to 0.47 | 7 | 0.15 0.06 to 0.31 | 1.86 | 0.18 |
| 10–14 | 3 173 226 | 1 654 964 | 1 518 262 | 28 | 0.22 0.15 to 0.32 | 16 | 0.24 0.14 to 0.39 | 12 | 0.20 0.10 to 0.35 | 1.22 | 0.60 |
| 15–19 | 3 438 414 | 1 826 179 | 1 612 235 | 123 | 0.89 0.74 to 1.07 | 92 | 1.26 1.02 to 1.54 | 31 | 0.48 0.33 to 0.68 | 2.62 | <0.05 |
| 20–24 | 3 055 420 | 1 625 371 | 1 430 049 | 151 | 1.24 1.05 to 1.45 | 100 | 1.54 1.25 to 1.87 | 51 | 0.89 0.66 to 1.17 | 1.73 | <0.05 |
| 25–29 | 3 538 949 | 1 802 805 | 1 736 144 | 152 | 1.07 0.91 to 1.26 | 84 | 1.16 0.93 to 1.44 | 68 | 0.98 0.76 to 1.24 | 1.18 | 0.29 |
| 30–34 | 3 695 348 | 1 866 397 | 1 828 951 | 209 | 1.41 1.23 to 1.62 | 112 | 1.50 1.24 to 1.81 | 97 | 1.33 1.08 to 1.62 | 1.13 | 0.37 |
| 35–39 | 4 099 147 | 2 060 233 | 2 038 914 | 239 | 1.46 1.28 to 1.65 | 129 | 1.57 1.31 to 1.86 | 110 | 1.34 1.10 to 1.61 | 1.17 | 0.25 |
| 40–44 | 4 131 423 | 2 071 431 | 2 059 992 | 307 | 1.86 1.66 to 2.08 | 162 | 1.96 1.67 to 2.28 | 145 | 1.76 1.48 to 2.07 | 1.11 | 0.36 |
| 45–49 | 4 073 358 | 2 044 641 | 2 028 717 | 304 | 1.87 1.66 to 2.09 | 148 | 1.81 1.53 to 2.13 | 156 | 1.92 1.63 to 2.25 | 0.94 | 0.60 |
| 50–54 | 3 798 131 | 1 887 973 | 1 910 158 | 405 | 2.67 2.41 to 2.94 | 204 | 2.70 2.34 to 3.10 | 201 | 2.63 2.28 to 3.02 | 1.03 | 0.79 |
| 55–59 | 2 766 695 | 1 360 747 | 1 405 948 | 327 | 2.95 2.64 to 3.29 | 150 | 2.76 2.33 to 3.23 | 177 | 3.15 2.70 to 3.65 | 0.88 | 0.23 |
| 60–64 | 2 182 236 | 1 057 035 | 1 125 201 | 283 | 3.24 2.88 to 3.64 | 127 | 3.00 2.50 to 3.57 | 156 | 3.47 2.94 to 4.05 | 0.86 | 0.23 |
| 65–69 | 1 812 168 | 833 242 | 978 926 | 236 | 3.26 2.85 to 3.70 | 85 | 2.55 2.04 to 3.15 | 151 | 3.86 3.27 to 4.52 | 0.66 | <0.05 |
| 70–74 | 1 566 014 | 672 894 | 893 120 | 178 | 2.84 2.44 to 3.29 | 72 | 2.68 2.09 to 3.37 | 106 | 2.97 2.43 to 3.59 | 0.90 | 0.50 |
| 75–79 | 1 084 367 | 410 726 | 673 641 | 108 | 2.49 2.04 to 3.01 | 49 | 2.98 2.21 to 3.94 | 59 | 2.19 1.67 to 2.82 | 1.36 | 0.11 |
| 80–84 | 595 509 | 186 008 | 409 501 | 56 | 2.35 1.78 to 3.05 | 13 | 1.75 0.93 to 2.99 | 43 | 2.63 1.90 to 3.54 | 0.67 | 0.20 |
| ≥85 | 366 609 | 94 736 | 271 873 | 13 | 0.89 0.47 to 1.52 | 6 | 1.58 0.58 to 3.45 | 7 | 0.64 0.26 to 1.33 | 2.45 | 0.09 |
| Total | 47 990 761 | 23 840 896 | 24 149 865 | 3144 | 1.64 1.58 to 1.70 | 1567 | 1.64 1.56 to 1.73 | 1577 | 1.63 1.55 to 1.72 | 1.01 | 0.86 |

The population of Korea was based on the 2010 census from the Korean Statistical Information Service. NA, not available.

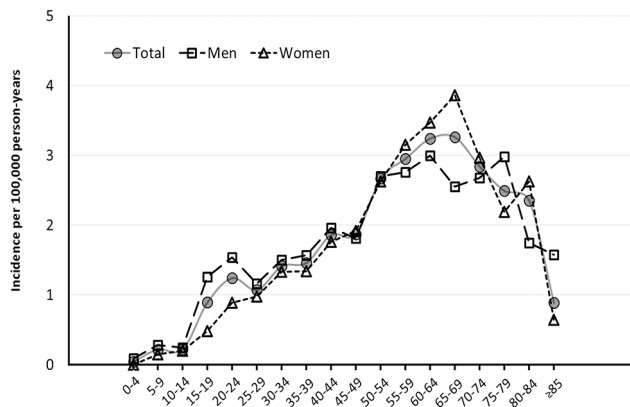

**Figure 1** Average incidence of retinitis pigmentosa (RP). Incidence per 100 000 person-years of RP in the South Korean population according to age groups from 2011 to 2014.

after age 69. There was a unique peak in incidence for men around 20 years of age (figure 1).

There were 851 (27.1%), 708 (22.5%), 728 (23.2%), and 857 (27.3%) incident cases in 2011, 2012, 2013, and 2014, respectively. There was little difference in the annual number of newly diagnosed patients with RP from 2011 to 2014. The annual incidence of RP was slightly higher in 2011 and 2014, and slightly lower in 2012 and 2013 (1.77 cases/100 000 population in 2011, 1.48 cases/100 000 population in 2012, 1.52 cases/100 000

population in 2013, and 1.79 cases/100 000 population in 2014) (table 2).

## DISCUSSION

In the present study, using the KNHIS database between 2011 and 2014, we found a nationwide, population-based, estimated RP incidence of 1.64 cases/100 000 person-years. This study involved a survey of the entire South Korean population using reliable data based on an ophthalmologist-confirmed RP registration.

RP is not an acute symptomatic disease like retinal vascular occlusion. The most common initial symptom of RP, night blindness, has an insidious onset and may go unperceived for many years. Patients often become cognisant of RP incidentally. Therefore, previous studies from Japan noted the disease onset as the time of a diagnosis of RP by an ophthalmologist.[9] Another study from Denmark discussed possible recall bias, and patients were classified as possible, probable, or certain according to the presence of specified diagnostic criteria and confirmatory electroretinographic changes. RP is very rare; therefore, it is impossible to estimate the incidence without a large cohort of subjects, such as that provided by nationwide data. However, it is practically impossible to perform ophthalmological examinations of the entire population to diagnose a rare intractable disease such as RP. Our study design was based on the ophthalmologist-confirmed RP registry, and it demonstrated a high specificity for the

**Table 2** Demographic characteristics of South Koreans and the annual incidence of retinitis pigmentosa per 100 000 person-years in the South Korean population (2011–2014)

| Age (years) | 2011 | | | 2012 | | | 2013 | | | 2014 | | |
|---|---|---|---|---|---|---|---|---|---|---|---|---|
| | Total | Men | Women | Total | Men | Women | Total | Men | Women | Total | Men | Women |
| 0–4 | 0.05 | 0.09 | 0.00 | 0.05 | 0.09 | 0.00 | 0.05 | 0.09 | 0.00 | 0.05 | 0.09 | 0.00 |
| 5–9 | 0.21 | 0.40 | 0.00 | 0.33 | 0.40 | 0.26 | 0.17 | 0.08 | 0.26 | 0.17 | 0.24 | 0.09 |
| 10–14 | 0.13 | 0.24 | 0.00 | 0.25 | 0.18 | 0.33 | 0.25 | 0.36 | 0.13 | 0.25 | 0.18 | 0.33 |
| 15–19 | 1.16 | 1.75 | 0.50 | 0.84 | 1.15 | 0.50 | 0.73 | 0.88 | 0.56 | 0.84 | 1.26 | 0.37 |
| 20–24 | 1.44 | 1.66 | 1.19 | 0.88 | 1.11 | 0.63 | 1.15 | 1.48 | 0.77 | 1.47 | 1.91 | 0.98 |
| 25–29 | 1.47 | 1.61 | 1.32 | 0.88 | 0.94 | 0.81 | 0.93 | 1.22 | 0.63 | 1.02 | 0.89 | 1.15 |
| 30–34 | 1.41 | 1.23 | 1.59 | 1.19 | 1.34 | 1.04 | 1.49 | 1.71 | 1.26 | 1.57 | 1.71 | 1.42 |
| 35–39 | 1.68 | 1.84 | 1.52 | 1.22 | 1.26 | 1.18 | 1.59 | 1.70 | 1.47 | 1.34 | 1.46 | 1.23 |
| 40–44 | 1.65 | 1.54 | 1.75 | 1.94 | 2.17 | 1.70 | 1.69 | 1.79 | 1.60 | 2.15 | 2.32 | 1.99 |
| 45–49 | 1.77 | 1.71 | 1.82 | 1.69 | 1.91 | 1.48 | 1.72 | 1.37 | 2.07 | 2.28 | 2.25 | 2.32 |
| 50–54 | 3.37 | 3.07 | 3.66 | 2.47 | 2.65 | 2.30 | 2.24 | 2.28 | 2.20 | 2.58 | 2.81 | 2.36 |
| 55–59 | 3.04 | 3.01 | 3.06 | 2.42 | 2.35 | 2.49 | 2.82 | 2.79 | 2.85 | 3.54 | 2.87 | 4.20 |
| 60–64 | 3.39 | 2.93 | 3.82 | 3.30 | 3.41 | 3.20 | 3.07 | 2.74 | 3.38 | 3.21 | 2.93 | 3.47 |
| 65–69 | 3.92 | 2.76 | 4.90 | 2.70 | 2.28 | 3.06 | 2.98 | 1.92 | 3.88 | 3.42 | 3.24 | 3.58 |
| 70–74 | 3.00 | 2.97 | 3.02 | 2.68 | 3.27 | 2.24 | 2.81 | 2.08 | 3.36 | 2.87 | 2.38 | 3.25 |
| 75–79 | 2.21 | 2.92 | 1.78 | 1.75 | 2.19 | 1.48 | 1.94 | 2.43 | 1.63 | 4.06 | 4.38 | 3.86 |
| 80–84 | 2.02 | 1.08 | 2.44 | 2.35 | 0.54 | 3.17 | 2.02 | 3.23 | 1.47 | 3.02 | 2.15 | 3.42 |
| ≥85 | 1.09 | 2.11 | 0.74 | 1.09 | 2.11 | 0.74 | 0.27 | 0.00 | 0.37 | 1.09 | 2.11 | 0.74 |
| Total | 1.77 | 1.74 | 1.81 | 1.48 | 1.56 | 1.40 | 1.52 | 1.50 | 1.53 | 1.79 | 1.77 | 1.80 |

diagnosis. Although not ideal, our approach was cost-effective for identifying patients with RP and calculating the incidence of RP in the entire South Korean population. Studying a large cohort, such as the entire South Korean population, and having a 4-year study period may provide stability to the heterogeneous detection rate seen with this insidious disease. These properties limit the chance probabilities that may occur with studies of smaller localised populations or tertiary hospital-based populations. Studies regarding the incidence of other rare intractable diseases, such as exudative age-related macular degeneration[7] and Moyamoya disease,[8] using the same registration database have been previously reported.

Using the South Korean national claims database from 2011 to 2014, which includes the RP registration programme, we estimated the incidence rate of RP as 1.64 cases/100 000 person-years (95% CI 1.58 to 1.70). To the best of our knowledge, only two previous studies have estimated the incidence of RP based on an adequate number of patients with RP. A well-defined local area study conducted in the 1980s in the US state of Maine estimated that the incidence of RP was 6 cases/100 000 person-years.[5] Another important previous study regarding the epidemiology of RP, performed in Denmark in the 2000s, reported an average incidence of 0.79 cases/100 000 person-years from 1990 to 1997.[4] The incidence of RP in South Korea found in the present study is lower than that seen in Maine and greater than that seen in Denmark.

Information regarding the age of disease onset is meaningful in genetic counselling, and a previous study from Japan described an age at onset curve for RP based on 370 patients with RP.[9] The age at onset, defined as the age of diagnosis, gradually increased with age until age 65 and subsequently remained steady after the age of 65.[9] In the Danish RP study,[4] the incidence of RP also increased until age 60–64 years, and then it decreased in people older than 65 years, similar to the trend seen in the present study (figure 1). In terms of sex, the Danish cohorts with RP showed a nearly equal increase in the annual incidence between men and women from 1990 to 1997.[4] The present study showed similar trends; the overall male-to-female ratio was 1.01 (p=0.86) and the annual incidence of RP was similar for men and women during the 4-year study period (table 2). However, the estimated age-specific incidence of newly diagnosed patients was nearly double in male patients compared with female patients in the 20–24-year-old age group (men 1.54 cases/100 000 population, women 0.89 cases/100 000 population) (table 1). In addition, the incidence of RP was greater in women than in men aged 50–65 years. In the Danish RP study, the age at onset was much higher in male patients than in female patients aged 6–18 years.[4] Our results also show that the onset may be earlier in male individuals than in female individuals. However, the specifics of military service in South Korea should also be considered. The higher incidence of RP in men in their 20s entering mandatory military service in South Korea may be because they are more frequently subjected to physical examinations related to potential exemption from the military service.

To the best of our knowledge, this is the largest population ever studied for this rare disease based on an ophthalmologist-confirmed diagnosis. It is also the first study to investigate the nationwide incidence of RP in an Asian country.

However, this study has several limitations. First, as aforementioned, it is difficult to define the exact time of onset of RP, and we arbitrarily defined the time of occurrence of RP as the time when the patients were diagnosed as having RP by an ophthalmologist, as in previous studies.[4][9] Second, we may have underestimated the incidence rates of RP. We identified patients with RP using healthcare claims but could not include asymptomatic patients or patients who did not use healthcare services. Since there is no cure for RP yet, some patients may not want to register, even though they have been diagnosed as having RP. However, it is reasonable to assume that people with symptomatic RP are likely to use healthcare services at some time in the course of the disease. The universal health insurance coverage and co-payment reduction with the registration programme for this rare and intractable disorder may also help to encourage healthcare use among patients with RP. Moreover, healthcare accessibility in South Korea is very high. The estimated incidence of newly diagnosed patients showed a consistent annual pattern, and the relatively long study period, including 4 years of data, may offset this limitation. Third, the diagnosis of RP was defined based on the healthcare registration and a Korean Classification of Diseases code, which may be inaccurate compared with a diagnosis obtained from a medical record. However, patients with RP in this study had their RP diagnoses confirmed by ophthalmologists based on the National Health Insurance Service diagnostic criteria, including electroretinography, which provides a test result that is considered a confirmatory criterion.[4] In addition, the eligibility of submitted registrations was reviewed comprehensively by the Health Insurance Review Agency. Fourth, a lack of clinical information, including family pedigree, visual acuity, visual field and genetic analysis, is another inherent limitation of the present claims database study. Fifth, we only targeted subjects in South Korea. Therefore, this result cannot be directly compared with those of other ethnic groups. Lastly, we did not report the prevalence of RP because of the characteristics of the South Korean national claims database. The analysis of prevalence for RP was impossible with the Korean Classification of Disease, sixth edition (Korean Classification of Diseases-6) system, in which RP, Stargardt disease, vitelliform retinal dystrophy and other unspecified hereditary retinal dystrophies were similarly categorised as hereditary retinal dystrophy (H35.5). In this study, we excluded subjects with RP registered before 31 December 2010 because it was difficult to distinguish between these diseases in the KNHIS database. In our explorative analysis, only 109 incident cases of Stargardt disease were

observed from 2011 to 2014, a relatively small number compared with the 3144 cases of RP. Therefore, we have presented the period prevalence of presumed RP in the Supplementary appendix as 15.47 cases/100 000 persons (~1/6500), a value consistent with previous reports (12–26 cases/100 000 persons or 1/3800–8300)[5 10–12] and slightly lower than that found in Danish cohorts with RP (22.4 cases/100 000 persons).[4]

In conclusion, we estimated the nationwide incidence of newly diagnosed patients with RP in South Korea using a database that included the entire national population during the 4-year period from 2011 to 2014. The estimated population-based incidence rate of RP for all ages was 1.64 cases/100 000 persons-years, and there was generally no difference in the rates between men and women. Our study should be helpful in assessing the RP-related socio-economic burden and in planning accordingly within the healthcare system.

**Acknowledgements** This work was supported by a National Health Insurance Ilsan Hospital grant (NHIMC 2015-02-015). This study used data from the NHIS-NCS 2002–2013 (NHIS-2015-1-068), which was released by the KNHIS. The authors alone are responsible for the content and writing of this article. Eun Jee Chung had full access to all of the data in the study, and takes responsibility for the integrity of the data and the accuracy of the data analysis.

**Contributors** Conception and design: THR and EJC. Analysis and interpretation: THR, HWP, DWK, and EJC. Data collection: THR, HWP, DWK, and EJC. Manuscript preparation: THR and HWP. Overall responsibility: EJC.

**Competing interests** None declared.

**Ethics approval** Institutional Review Board of the National Health Insurance Ilsan Hospital.

**Provenance and peer review** Not commissioned; externally peer reviewed.

**Data sharing statement** Access to NHIS-NSC data are available from the website of (NHIS) after completing the application process and receiving approval. Detailed cohort profile and the methods for obtaining data are explained in the following source: Lee J, Lee JS, Park SH, Shin SA, Kim K. Cohort Profile: The National Health Insurance Service-National Sample Cohort (NHIS-NSC), South Korea. International journal of epidemiology. 2016. doi: 10.1093/ije/dyv319. PubMed PMID: 26822938.

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
