## [Reviewer comments · BMJ Open]

ARTICLE DETAILS

TITLE (PROVISIONAL)	Four-Year Nationwide Incidence of Retinitis Pigmentosa in South Korea: A Population-based Retrospective Study from 2011 to 2014
AUTHORS	Rim, Tyler Hyung Taek; Park, Hye Won; Kim, Dong Wook; Chung, Eun Jee

VERSION 1 - REVIEW

REVIEWER	Tsai, Der-Chong National Yang-Ming University Hospital, Taiwan.
REVIEW RETURNED	04-Jan-2017

GENERAL COMMENTS	In this nationwide population-based retrospective study, authors reported the incidence of RP in South Korea using the KNHIS database. It is the first report among Asian population. The study design and methodology of the current study were reasonable. Only minor comments. 1. The denominator (total person-time at risk) was assumed as the total population (Census in 2010, N=47 990 761) over 4 years from 2011 to 2014 (population in 2010 multiplied by four) in this study. Although it was not exactly accurate for the incidence calculation but it was an acceptable approximation.2. The age group definition. Were the age groups classified according to the age in 2010?3. The column subheading of Table S1 should be "period prevalence" rather than "incidence".
--

REVIEWER	Francesco Saverio Sorrentino Ospedale Maggiore Bologna - ITALY
REVIEW RETURNED	13-Feb-2017

GENERAL COMMENTS	Good analysis. Interesting topic. Well-organized full-article.
--

REVIEWER	Javier Ruiz-Ederra Biodonostia Research Institute, Spain
REVIEW RETURNED	16-Feb-2017

GENERAL COMMENTS	Taking advantage of the universal health insurance system of South Korea, the authors have performed a comprehensive study of the incidence of retinitis pigmentosa based on the whole South Korean population. Since this is one of the few works including data from a whole large country, with about 48 million inhabitants, it is of the general interest for ophthalmologists as well as for retina
---

	researchers. I have only few minor suggestions that would need to be addressed: 1: One of the main limitations of the present study is the lack of information regarding molecular diagnosis of the population included in the study. Although the inclusion criteria of the patients was based on clinical diagnosis it is highly likely that at least a portion of the patients have a molecular/genetic diagnostic report. Taking into account the recent advancements and the implementation of Next Generation Sequencing and other procedures for genetic disorders in general, with retinitis pigmentosa as one of the most relevant examples, it is highly likely that many of the RP patients have information regarding the genetic cause of their disease. Therefore I would strongly suggest incorporating this information to the manuscript if available. (i.e. the proportion of cases with known mutations; and a distribution of those genes/mutations more prevalent etc..). 2: I would recommend to include prevalence data not only referred to “cases in 100.000 people”, but also to 1 case per xxxx people, which typically has been reported to be around 1 in 3500-4000 cases worldwide. 3: Despite the manuscript is well written in general, there are few instances where it might benefit for an editing of the English employed. An example of this include: -Page 9, lines 19-21: “The incidence or RP..... is less than...” Consider replacing less here.
--	---

VERSION 1 – AUTHOR RESPONSE

Reviewer: 1

Reviewer Name: Tsai, Der-Chong

Institution and Country: National Yang-Ming University Hospital, Taiwan.

Competing Interests: none declared

In this nationwide population-based retrospective study, authors reported the incidence of RP in South Korea using the KNHIS database. It is the first report among Asian population.

The study design and methodology of the current study were reasonable.

Only minor comments.

Comment 1: The denominator (total person-time at risk) was assumed as the total population (Census in 2010, N=47 990 761) over 4 years from 2011 to 2014 (population in 2010 multiplied by four) in this study. Although it was not exactly accurate for the incidence calculation but it was an acceptable approximation.

Response: We agree that although the denominator is not exact, it is a good approximation for a rare disease such as retinitis pigmentosa (RP).

Comment 2: The age group definition. Were the age groups classified according to the age in 2010?

Response: Per your request, we have included a definition of the age group as follows.

(Page 6, lines 123: ‘The annual incidence from 2011 to 2014 and overall incidence were estimated with 95% confidence intervals; additionally, the age-specific and sex-specific incidence rates were estimated, and the age-specific population for each year was approximated based on the 2010 census population.’

Comment 3: The column subheading of Table S1 should be “period prevalence” rather than “incidence”.

Response: Per your recommendation, we have corrected the subheading of Table S1.

Reviewer: 2

Reviewer Name: Francesco Saverio Sorrentino

Institution and Country: Ospedale Maggiore Bologna - ITALY

Competing Interests: None declared

Comment 1: Good analysis. Interesting topic. Well-organized full-article.

Response: Thank you for your favourable review and positive feedback.

Reviewer: 3

Reviewer Name: Javier Ruiz-Ederra

Institution and Country: Biodonostia Research Institute, Spain

Competing Interests: none declared

Taking advantage of the universal health insurance system of South Korea, the authors have performed a comprehensive study of the incidence of retinitis pigmentosa based on the whole South Korean population. Since this is one of the few works including data from a whole large country, with about 48 million inhabitants, it is of the general interest for ophthalmologists as well as for retina researchers.

I have only few minor suggestions that would need to be addressed:

Comment 1: One of the main limitations of the present study is the lack of information regarding molecular diagnosis of the population included in the study. Although the inclusion criteria of the patients was based on clinical diagnosis it is highly likely that at least a portion of the patients have a molecular/genetic diagnostic report. Taking into account the recent advancements and the implementation of Next Generation Sequencing and other procedures for genetic disorders in general, with retinitis pigmentosa as one of the most relevant examples, it is highly likely that many of the RP patients have information regarding the genetic cause of their disease.

Therefore I would strongly suggest incorporating this information to the manuscript if available. (i.e. the proportion of cases with known mutations; and a distribution of those genes/mutations more prevalent etc.).

Response: We agree that the genomic approach for patients with RP is very important. Unfortunately, our registry database did not include detailed information about RP for each individual. Therefore, we have added this study limitation to the paragraph on limitations in the Discussion section.

\ Page 6, lines 231: ‘Fourth, a lack of clinical information, including family pedigree, visual acuity, visual field, and genetic analysis, is another inherent limitation of the present claims database study.’

Comment 2: I would recommend to include prevalence data not only referred to “cases in 100.000 people”, but also to 1 case per xxxx people, which typically has been reported to be around 1 in 3500-4000 cases worldwide.

Response: Per your suggestion, we have provided the prevalence data in 1/6 500 case/persons format in the manuscript and appendix.

\ Page 12, lines 243: ‘Therefore, we have presented the period prevalence of presumed RP in the supplementary digital content 1 of the appendix as 15.47 cases/100 000 persons (~1/6500),...’

Comment 3: Despite the manuscript is well written in general, there are few instances where it might benefit for an editing of the English employed. An example of this include:

-Page 9, lines 19-21: “The incidence or RP..... is less than...” Consider replacing less here.

Response: Per your recommendation, we hired an English language editing service to proofread our revised manuscript.

VERSION 2 – REVIEW

REVIEWER	Tsai Der-Chong National Yang-Ming University Hospital
REVIEW RETURNED	23-Mar-2017

GENERAL COMMENTS	My concerns have been adequately addressed in the revised manuscript.
---

REVIEWER	Javier Ruiz Ederra Biodonostia Health Research Institute, Spain
REVIEW RETURNED	13-Mar-2017

GENERAL COMMENTS	I have seen that all issues raised have been satisfactorily addressed, and therefore I consider it adequate for publication.
--